# Genetic Diversity of a Rising Invasive Pest in the Native Range: Population Genetic Structure of *Aromia bungii* (Coleoptera: Cerambycidae) in South Korea

**Seunghyun Lee** [1], **Deokjea Cha** [2], **Yongwoo Nam** [3] **and Jongkook Jung** [3,4,*]

1 Key Laboratory of Zoological Systematics and Evolution, Institute of Zoology, Chinese Academy of Sciences, 92 Box, No. 1 Beichen West Road, Chaoyang District, Beijing 100101, China; chiyark@snu.ac.kr
2 Restoration Assessment Team, Research Center for Endangered Species, National Institute of Ecology, Yeongyang 36531, Korea; developcha@gmail.com
3 Forest Insect Pests and Diseases, National Institute of Forest Science, Seoul 02455, Korea; orangmania99@korea.kr
4 Department of Forest Environment Protection, Kangwon National University, Chuncheon 24341, Korea
* Correspondence: jkjung@kangwon.ac.kr

**Abstract:** The red-necked longhorn beetle (RLB; *Aromia bungii* [Faldermann, 1835]) is an emerging invasive pest. From its native range of East Asia, it invaded Europe and Japan in the early 2010s. Despite its increasing importance, the molecular resources of RLB are scarce, and its invasive dynamics are largely unknown. In the present study, we carried out the first analysis of its population genetic structure in South Korea, which is part of its native range, using 1248 bp cytochrome oxidase subunit I (COI) sequences of 199 individuals from 18 localities. We found that in South Korea, RLB has a moderate population genetic structure and can be divided into three geographical subgroups: central, southeastern, and southwestern subgroup. Comparative analyses with two Chinese, one German, and ten Italian RLB sequences yielded non-significant results because of largely missing genetic data from other native areas. Nevertheless, as it provided the first population genetic data for this invasive alien species (IAS) whose range is increasing, our research is a crucial molecular resource for future invasive dynamics research.

**Keywords:** red-necked longhorn beetle; *Aromia bungii*; population genetics; COI; invasive species

## 1. Introduction

Invasive alien species (IAS) are introduced into non-native areas accidentally or intentionally through anthropogenic activities, posing threats for the environment they invade [1]. Insects are among the more serious IAS, occupying 25% of the 100 of the world's worst invasive alien species [2]. The damage from invasive insects has sharply increased [3] with the increase in global transportation and logistics; moreover, new invasive insects are being discovered more frequently than ever before [4].

One of the recently recognized and emerging IAS is *Aromia bungii* (Faldermann, 1835) (Coleoptera: Cerambycidae), also known as the red-necked longhorn beetle (RLB). RLB is an important pest species in agricultural landscapes, damaging various stone fruit trees (the genus *Prunus*), including peach, apricot, plum, and cherry [5]. In recent years, RLB arose as a serious pest in South Korea, and is a threat to the famous cherry blossom trees (*Prunus yedoensis* Matsum., 1901) in the urban landscape, weakening and killing them [6,7]. The genus *Prunus* is the main host but RLB can also utilize various plant species belonging to the families *Fagaceae*, *Juglandaceae*, *Meliaceae*, *Poaceae*, *Punicaceae*, *Rosaceae*, *Rutaceae*, *Salicaceae*, and *Theaceae* as host plants, according to an unverified report [8]. Their early stage larvae bore under the bark of living or weakened trees, feeding on phloem and cambium, leaving slight impressions on the sapwood [5]. The mature larvae bore heartwood and eventually weaken and kill the host trees [6].

RLB is native to East Asia including China [9], Korea [10], Mongolia [11], Taiwan [12], and Vietnam [13]. RLB began to be considered an invasive pest after it invaded Europe and Japan in the early 2010s. Detailed reviews of its invasion history and past outbreaks were carried out for Europe [5,14] and for Japan [15,16]. Along with the economic importance of RLB and the growing records of interceptions in quarantine [10], its invasive dynamics are becoming increasingly important. Global invasion and interception history are graphically summarized in Figure 1D.

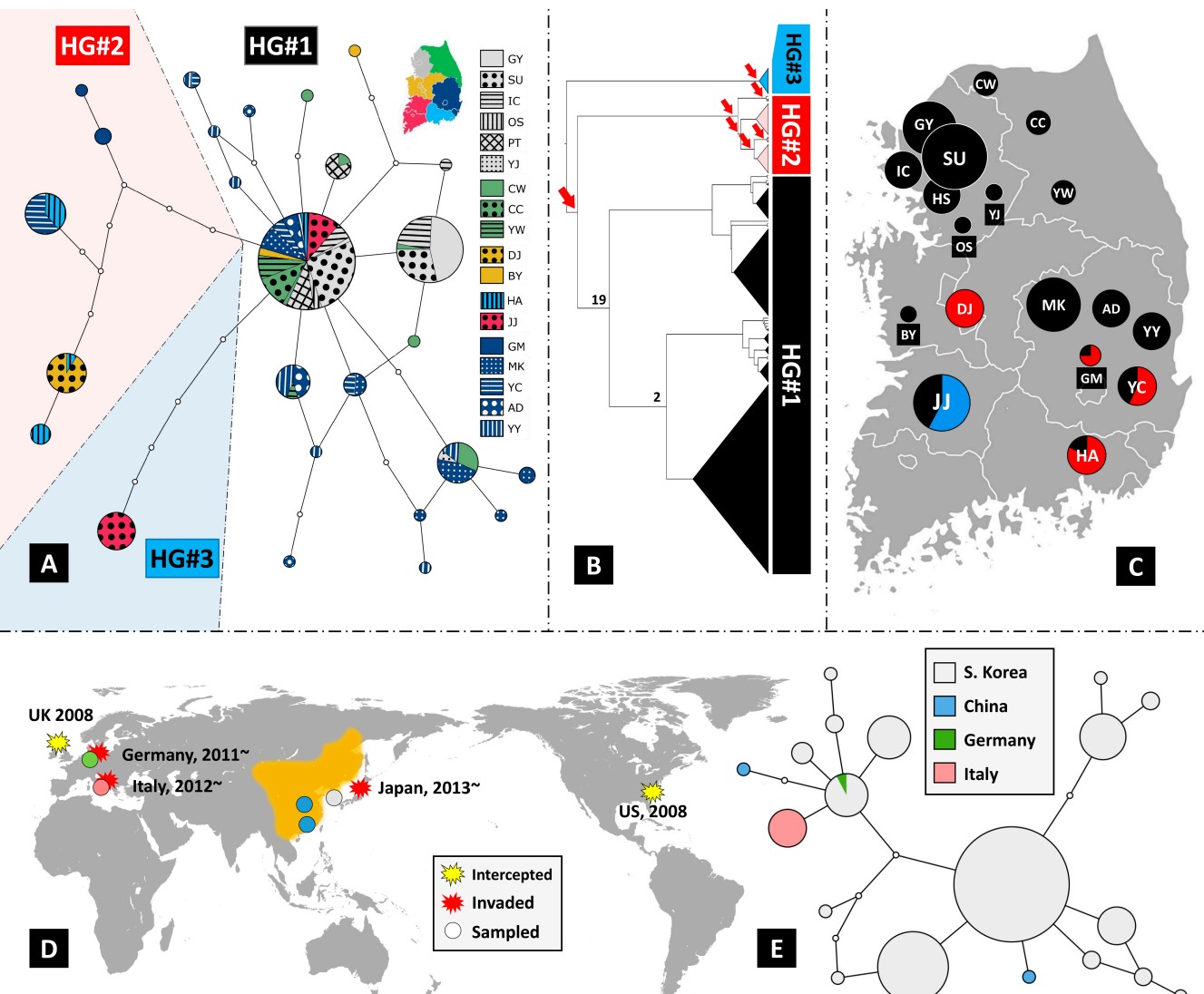

**Figure 1.** (**A**) Genetic relationships using K-dataset. K-dataset was divided into three haplogroups indicated as HG1~3; (**B**) Phylogenetic relationship using K-dataset. High Ultrafast bootstrap support values (>90) are marked with red arrows. Numbers in two nodes indicate Ultrafast bootstrap values. Colors of the haplogroups in pie chart correspond to Figure 1A; (**C**) Haplogroup distribution of K-dataset and their genetic clusters inferred from *COI* data. Colors of the haplogroups in pie chart correspond to Figure 1A; (**D**) Orange mark: global distribution of RLB. Circles: sampled sites of G-dataset. Red explosion icon: invaded areas. Yellow explosion icon: intercepted areas. (**E**) Genetic relationships using K-dataset. Labels of collection sites are indicated in the upper right boxes.

Despite the growing importance of RLB invasion, its invasive dynamics have not been explored; only a preliminary comparison between 10 European and two Chinese cytochrome oxidase subunit I (COI) sequences has been published to date [5]. To infer the invasion history of this IAS, it is necessary to analyze the genetic structure of its native and invasive populations [17]. However, only a limited number of sequences are available in

the Genbank database: one mitochondrial genome, 16 COI, 14 large subunit ribosomal RNA gene (28s rDNA), and 10 other minor loci (e.g., 12s rDNA, 16s rDNA). Among them, one mitochondrial genome sequence and two COI sequences were obtained from individuals collected in their native areas, whereas most sequences (ten COI and ten 28s rDNA) were obtained from individuals collected in Europe, i.e., the invaded area [5,18]. Therefore, the most important issue in RLB invasive dynamics is sequencing the beetles from their native range and investigating their genetic structure.

In the present study, we sequenced partial mitochondrial DNA sequences of RLB from various regions of South Korea and studied their genetic structure. By providing the first comprehensive sequence from its native population, this research would be the first stepping stone for future RLB invasive dynamics studies.

## 2. Materials and Methods

Two data sets were used for the analysis: a long-sequence dataset that included only Korean samples (K-dataset) and a shorter dataset containing Korean samples and some sequences from other countries (G-dataset).

We collected 199 DNA-grade RLB samples by hand from 18 localities in South Korea (Table 1). Total genomic DNA was extracted from the thoracic muscles of adult individuals, abdominal segments of larvae, and whole eggs (see Table 1). The DNeasy Blood and Tissue Kit (Qiagen, Hilden, Germany) was used for DNA extraction according to the manufacturer's protocol. All samples were preserved in DNA-grade ethanol at −20 °C for future analyses.

**Table 1.** RLB samples included in this study. Collected locality, abbreviation (Abb), province (Prov.), GPS data, number of samples used in genetic analyses (No.), larvae (L), egg (E), collection date (Coll.), and Genbank accession numbers.

| | Locality | Abb | Prov | GPS-N | GPS-E | No. | Coll. | Genbank Accession |
|---|---|---|---|---|---|---|---|---|
| 1 | Goyang | GY | GG | 37.65 | 126.89 | 17 | 2019.08.08 | OK428971~OK428987 |
| 2 | Hwaseong | PT | GG | 37.16 | 126.90 | 11 | 2020.07.15 | OK429054~OK429064 |
| 3 | Incheon | IC | GG | 37.43 | 126.71 | 13 | 2020.07.08 | OK428999~OK429011 |
| 4 | Osan | OS | GG | 37.14 | 127.08 | 2 | 2020.07.15 | OK429052~OK429053 |
| 5 | Seoul-NS | SU | GG | 37.55 | 126.98 | 4 | 2020.07.09 | OK429048~OK429051 |
| 6 | Seoul-YED | SU | GG | 37.53 | 126.91 | 14 | 2020.07.09 | OK429091~OK429104 |
| 7 | Seoul-YED | SU | GG | 37.53 | 126.91 | 12 | 2019.09.23 | OK429065~OK429076 |
| 8 | Yeoju | YJ | GG | 37.24 | 127.65 | 1 (1 E) | 2020.07.15 | OK429105 |
| 9 | Buyeo | BY | CN | 36.28 | 126.91 | 3 (2 L) | 2020.07.10 | OK428936~OK428938 |
| 10 | Daejeon | DJ | CN | 36.32 | 127.43 | 11(1 L) | 2020.07.08 | OK428956~OK428966 |
| 11 | Cheorwon | CW | GW | 38.14 | 127.30 | 9 | 2020.07.07 | OK428947~OK428955 |
| 12 | Chuncheon | CC | GW | 37.92 | 127.78 | 8 | 2020.07.21 | OK428939~OK428946 |
| 13 | Yeongwol | YW | GW | 37.18 | 128.47 | 7 | 2020.07.05–08 | OK429106~OK429112 |
| 14 | Jeonju-BJW | JJ | JB | 35.85 | 127.14 | 9 | 2020.07.22 | OK429012~OK429020 |
| 15 | Jeonju-YYE | JJ | JB | 35.84 | 127.14 | 10 | 2020.07.21 | OK429021~OK429030 |
| 16 | Andong | AD | GB | 36.57 | 128.76 | 10 | 2020.07.05–08 | OK428926~OK428935 |
| 17 | Gumi | GM | GB | 36.10 | 128.39 | 4 | 2020.08.20 | OK428967~OK428970 |
| 18 | Mungyeong | MK | GB | 36.59 | 128.19 | 17 | 2020.07.21 | OK429031~OK429047 |
| 19 | Yeongcheon | YC | GB | 35.97 | 128.91 | 14 | 2020.08.20 | OK429077~OK429090 |
| 20 | Yeongyang | YY | GB | 36.66 | 129.12 | 12 | 2020.08.20 | OK429113~OK429124 |
| 21 | Haman | HA | GN | 35.36 | 128.48 | 11 (10 L) | 2019.09.26 | OK428988~OK428998 |

Mitochondrial COI, which has been widely used in population genetics studies, was targeted and sequenced using Sanger's method in the present study. Four new RLB-specific primer sets were designed using the longest 1545 bp COI sequence from NCBI (accession number: KF737790.1): ABC_F1 (GGA ATA GTA GGA ACT TCT TTG AG 79-101), ABC_R2 (AAT TGG CAG TTC TGA GTA TCT ATG), ABC_R3 (CTA CAG TAA ATA TGT GAT GAG CTC), and ABC_F4 (AGA AGC CTT TGG CAC TCT CG). Most of the samples were successfully amplified using a single primer set, either ABC_F1 or ABC_R2. Some degraded samples were amplified using two primer sets, either ABC_F1 and ABC_R3, or ABC_F4

and ABC_R2. The Accupower PCR Premix (Bioneer, Daejeon, Korea) was used for PCR amplification. The thermal conditions were as follows: 95 °C for 15 min, followed by 40 cycles of denaturation at 95 °C for 1 min, annealing at 55–60 °C for 1 min, and extension at 72 °C for 1 min, with a final elongation at 72 °C for 4 min. PCR amplicons were purified and sequenced by Bionics Inc. (Seoul, Korea). MAFFT [19] was used for multiple sequence alignment, and the amino acid translation option in MEGA 7 [20] was used for the final sequence assessment.

To study the K-dataset, phylogenetic, network, and population genetic analyses were carried out. A phylogenetic tree was reconstructed using the maximum likelihood (ML) method on the IQ-TREE web server [21]. The best-fitting substitution model (TPM2 + F + G4) was determined using Modelfinder [22] under the Bayesian information criterion. The ultrafast bootstrap nodal support value was evaluated using 1000 replicates. Genetic relationships were inferred by TCS2.0 [23] using the network estimation approach under a 95% probability criterion for a parsimonious connection and visualized using tcsBU [24]. Arlequin 3.562 [25] was used to calculate the number of haplotypes, haplotype diversity, nucleotide diversity, pairwise FST values, and Tajima's D (TD).

To study the G-dataset, only network analysis was carried out, and the genetic relationship was visualized using the abovementioned method. One German (KM443233), two Chinese (MT371041, KF737790), and ten Italian (MN662926-MN662935) sequences were retrieved, aligned with the Korean dataset, and analyzed [18]. As only a few non-Korean RLB sequences were available at this point, the other analyses would not be significant.

## 3. Results and Discussion

The K-dataset comprised 1248 bp *COI* sequences in 199 RLB samples without any gaps or ambiguities in the final matrix. The GenBank accession numbers are listed in Table 1. In total, 26 haplotypes were observed in South Korea. The number of haplotypes (Nh), genetic diversity (Gd), nucleotide diversity (Nd), Tajima's D (TD), and *p*-value of TD (TDp) are shown in Table 2. Pairwise FST values and corresponding *p*-values are provided in Table 3.

**Table 2.** Collected locality, abbreviation, number of samples (N), number of haplotypes (Nh), genetic diversity (Gd), nucleotide diversity (Nd), Tajima's D (TD), and *p*-value of TD (TDp) based on 1248 bp dataset.

|  | Locality | Abbreviation | N | Nh | Gd | Nd | TD | TDp |
|---|---|---|---|---|---|---|---|---|
| 1 | Goyang | GY | 17 | 2 | $0.1176 \pm 0.1012$ | $0.000094 \pm 0.000175$ | −1.164 | 0.140 |
| 2 | Hwaseong | PT | 11 | 2 | $0.5091 \pm 0.1008$ | $0.000408 \pm 0.000420$ | 1.186 | 0.879 |
| 3 | Incheon | IC | 13 | 3 | $0.5641 \pm 0.1117$ | $0.000493 \pm 0.000467$ | −0.127 | 0.420 |
| 4 | Osan | OS | 2 | 1 | $0.0000 \pm 0.0000$ | $0.000000 \pm 0.000000$ | 0.000 | 1.000 |
| 5 | Seoul | SU | 30 | 2 | $0.4598 \pm 0.0612$ | $0.000368 \pm 0.000371$ | 1.280 | 0.910 |
| 6 | Yeoju | YJ | 1 | 1 | $1.0000 \pm 0.0000$ | $0.000000 \pm 0.000000$ | 0.000 | 1.000 |
| 7 | Buyeo | BY | 3 | 2 | $0.6667 \pm 0.3143$ | $0.001603 \pm 0.001511$ | 0.000 | 0.894 |
| 8 | Daejeon | DJ | 11 | 1 | $0.0000 \pm 0.0000$ | $0.000000 \pm 0.000000$ | 0.000 | 1.000 |
| 9 | Cheorwon | CW | 9 | 6 | $0.8333 \pm 0.1265$ | $0.001914 \pm 0.001300$ | −0.323 | 0.402 |
| 10 | Chuncheon | CC | 8 | 1 | $0.0000 \pm 0.0000$ | $0.000000 \pm 0.000000$ | 0.000 | 1.000 |
| 11 | Yeongwol | YW | 7 | 2 | $0.2857 \pm 0.1964$ | $0.000229 \pm 0.000313$ | −1.006 | 0.254 |
| 12 | Jeonju | JJ | 19 | 2 | $0.5146 \pm 0.0517$ | $0.002062 \pm 0.001292$ | 2.484 | 0.999 |
| 13 | Andong | AD | 10 | 4 | $0.7333 \pm 0.1005$ | $0.001211 \pm 0.000897$ | −0.582 | 0.310 |
| 14 | Gumi | GM | 4 | 3 | $0.8333 \pm 0.2224$ | $0.002003 \pm 0.001613$ | −0.797 | 0.182 |
| 15 | Mungyeong | MK | 17 | 6 | $0.8015 \pm 0.0646$ | $0.001343 \pm 0.000927$ | 0.431 | 0.697 |
| 16 | Yeongcheon | YC | 14 | 4 | $0.6484 \pm 0.1163$ | $0.003513 \pm 0.002072$ | 1.055 | 0.880 |
| 17 | Yeongyang | YY | 12 | 8 | $0.8939 \pm 0.0777$ | $0.001967 \pm 0.001287$ | 0.898 | 0.817 |
| 18 | Haman | HA | 11 | 4 | $0.7455 \pm 0.0978$ | $0.003351 \pm 0.002034$ | 0.957 | 0.855 |

**Table 3.** Pairwise FST calculated by Arlequin based on 1248 bp of COI sequence dataset (* = *p*-value < 0.05).

|  | AD | BY | CC | CW | DJ | GM | GY | HA | IC | JJ | MK | SU | OS | PT | YC | YY | YW |
|---|---|---|---|---|---|---|---|---|---|---|---|---|---|---|---|---|---|
| AD |  |  |  |  |  |  |  |  |  |  |  |  |  |  |  |  |  |
| BY | 0.161 |  |  |  |  |  |  |  |  |  |  |  |  |  |  |  |  |
| CC | 0.209 * | 0.342 |  |  |  |  |  |  |  |  |  |  |  |  |  |  |  |
| CW | 0.234 * | 0.119 | 0.212 |  |  |  |  |  |  |  |  |  |  |  |  |  |  |
| DJ | 0.908 * | 0.952 * | 1 * | 0.873 * |  |  |  |  |  |  |  |  |  |  |  |  |  |
| GM | 0.541 * | 0.391 | 0.743 * | 0.495 * | 0.891 * |  |  |  |  |  |  |  |  |  |  |  |  |
| GY | 0.651 * | 0.779 * | 0.915 * | 0.503 * | 0.991 * | 0.863 * |  |  |  |  |  |  |  |  |  |  |  |
| HA | 0.577 * | 0.446 | 0.609 * | 0.551 * | 0.378 * | 0.354 | 0.743 * |  |  |  |  |  |  |  |  |  |  |
| IC | 0.418 * | 0.425 * | 0.534 * | 0.292 * | 0.957 * | 0.726 * | 0.121 | 0.663 * |  |  |  |  |  |  |  |  |  |
| JJ | 0.451 * | 0.377 * | 0.449 * | 0.438 * | 0.817 * | 0.584 * | 0.637 * | 0.594 * | 0.533 * |  |  |  |  |  |  |  |  |
| MK | 0.357 * | 0.266 | 0.352 * | −0.007 | 0.878 * | 0.59 * | 0.637 * | 0.628 * | 0.481 * | 0.516 * |  |  |  |  |  |  |  |
| SU | 0.351 * | 0.379 | 0.181 | 0.31 * | 0.954 * | 0.782 * | 0.511 * | 0.745 * | 0.175 * | 0.577 * | 0.476 * |  |  |  |  |  |  |
| OS | −0.069 | −0.2 | 0 | −0.079 | 1 * | 0.462 | 0.885 | 0.439 | 0.361 * | 0.291 | 0.16 | −0.017 |  |  |  |  |  |
| PT | 0.267 * | 0.287 | 0.25 | 0.233 * | 0.964 * | 0.702 * | 0.788 * | 0.613 * | 0.501 * | 0.477 * | 0.389 * | 0.312 * | −0.01 |  |  |  |  |
| YC | 0.398 * | 0.263 | 0.417 * | 0.397 * | 0.579 * | 0.255 | 0.602 * | 0.077 | 0.504 * | 0.484 * | 0.482 * | 0.583 * | 0.221 | 0.429 * |  |  |  |
| YY | 0.036 | 0.137 | 0.204 * | 0.116 * | 0.852 * | 0.489 * | 0.558* | 0.571 * | 0.378 * | 0.447 * | 0.198 * | 0.371 * | −0.054 | 0.261 * | 0.407 * |  |  |
| YW | 0.067 | 0.205 | 0.02 | 0.179 * | 0.985 * | 0.676 * | 0.845 * | 0.583 * | 0.471 * | 0.429 * | 0.327 * | 0.179 | −0.313 | 0.195 | 0.392 * | 0.119 |  |
| YJ | 0.496 | 0.333 | 1 | −0.344 | 1 | 0.524 | 0.96 | 0.477 | 0.778 | 0.474 | −0.425 | 0.803 | 1 | 0.785 | 0.31 | 0.08 | 0.867 |

The haplotype network result, maximum-likelihood proportional tree inferred from IQ-TREE, and haplotype diversity of each location in South Korea are graphically summarized in Figure 1A–C. Our network analysis yielded a result similar to the results of the phylogenetic analysis, with the 26 haplotypes divided into three distinct clusters. Three distinct clades were observed in the ML tree, with high supporting values (UFboostrap > 90, indicated by arrows in Figure 1B) at most of the stem nodes. Regarding the consensus of phylogenetic and network analyses, the Korean RLB samples were divided into three haplogroups, namely HG1–HG3. HG1 was the most abundant haplogroup in South Korea, accounting for 78.9% (157/199) of the total sample. Among all haplotypes, haplotype A occurred in all six provinces of Korea and was the most abundant, accounting for 37.6% (75/199) of the total samples. In all samples from the north to the central region of South Korea, only HG1 was recorded, except for DJ, which only included HG2. HG2 was the second most abundant haplogroup, accounting for 15.6% of the total samples, and was the dominant haplogroup in the southeastern region of South Korea (75% in GM, 81.8% in HA, and 57.1% in YC). HG3 accounted for 5.5% of the total samples and was locally observed only in JJ. As our study did not include samples from Jeolla Province (southwestern part of South Korea, see red region in the map of Figure 1A), the potential distribution of HG3 might be wider than indicated by our results. We believe that determining the extent of the distribution of HG3 will be a crucial factor for describing the RLB population structure in South Korea and that this distribution should be investigated in the future. Our analyses revealed moderate genetic diversity (Gd in Table 2) and nucleotide diversity (Nd in Table 2) of South Korean RLB. Three localities (OS, DJ, and CC) had no Gd and Nd, whereas high to moderate levels of Gd were observed in the other localities. YY showed the highest Gd and haplotype number (Nh) (Gd = 0.8939 ± 0.0777, Nh = 8), followed by GM (Gd = 0.8333 ± 0.2224, Nh = 3) and CW (Gd = 0.8333 ± 0.1265, Nh = 6). The Gd of YJ was high because only one sample was included in the analysis. Tajima's Ds (abbreviated as TD in Table 2) of all localities were not significant (*p* > 0.05).

Despite population genetic structure can differ by species [26], sampling [27], and the selected genetic marker [28], our result was different from those of the two closely related species that shows nonstar-like network topology: Chinese *Neocerambyx raddei* Blessig, 1872 [28], Korean and global *Anoplophora glabripennis* (Motschulsky, 1854) [17]. The haplotype network result of Korea RLB showed a moderate star-like topology that many minor haplotypes are related to a major central haplotype. In comparison to our result that the major haplogroup and major haplotype (HG1) are observed in most of the sampled locality, both studies [17,28] rather showed all the sampled populations showed a lack of shared haplotypes. The ubiquitous distribution of major haplogroups (HG1), geographically separated two minor haplogroups (HG2 and HG3), and the moderate overall genetic structure indicates RLB populations in Korea might not be influenced by strong anthropogenic events. Recently, the invasion of non-native Asian longhorn beetle (*A. glabripennis*) in a conspecific native range was reported based on the recent narrow genetic diversity of two putative invasive subgroups, and overall population genetic structure, and other biological evidence [11]. However, our molecular result showed no evidence of RLB invasion in South Korea in that they have moderate genetic diversity and continuous population genetic structure. In South Korea, RLB was most frequently found on extremely large *P. yedoensis* individuals planted along roadsides in urban landscapes, rather than on peach and plum trees planted in orchards ([6], personal observation). Even if the invasion had occurred, the invaded populations would have been difficult to settle in considering their specific host preference in terms of host size and conditions.

The additional G-dataset comprised 479 bp *COI* sequences in 212 RLB samples without any gaps or ambiguities in the final matrix. A total of 19 haplotypes were identified. Due to sampling bias, the South Korean RLB showed the highest haplotype diversity (Nh = 16). Two Chinese samples collected from remote provinces showed a distinct genetic distance (Figure 1E). As RLB is also widely distributed across China, its genetic diversity within the country must be higher than that in South Korea. In a previous study, ten Italian RLB sequences were completely identical, indicating a possible bottleneck of the invasive population, even though all the samples were collected in geographically close localities [5]. In the present study, the only sample from Germany had the same haplotype as that of the Korean sample (Figure 1E), and we were not able to create any invasion scenario under the current sampling conditions because population genetic studies in the native range of RLB, including China, Vietnam, and Taiwan, are necessary for in advance to unveil its invasion dynamics.

**Author Contributions:** Conceptualization, S.L., D.C., Y.N. and J.J.; Data curation, S.L., D.C. and J.J.; Formal analysis, S.L. and D.C.; Investigation, S.L., D.C. and J.J.; Methodology, S.L.; Resources, S.L. and J.J.; Supervision, Y.N. and J.J.; Visualization, S.L.; Writing—original draft, S.L.; Writing—review and editing, Y.N. and J.J. All authors have read and agreed to the published version of the manuscript.

**Funding:** This study was supported by the National Institute of Forest Science (Project no. FE0100-2018-01).

**Institutional Review Board Statement:** Not applicable.

**Data Availability Statement:** All the COI sequences used in this study are deposited in Genbank. Genbank accession numbers are provided in Table 1.

**Acknowledgments:** The first author would like to thank Hyunkyu Jang, Woong Choi, Yo-eun Yang, Jinwook Park, Minsoo Dong, and SNU insect biosystematics lab members for their cooperation on taxon sampling.

**Conflicts of Interest:** The authors declare no conflict of interest.

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
