# Peer review of "Genetic Diversity of a Rising Invasive Pest in the Native Range: Population Genetic Structure of Aromia bungii (Coleoptera: Cerambycidae) in South Korea"

_diversity, doi:10.3390/d13110582_

Round 1

Reviewer 1 Report

I read with attention the manuscript entitled Genetic diversity of a rising invasive pest in the native range: population genetic structure of Aromia bungii (Coleoptera: Cerambycidae) in South Korea. Manuscripts deals with the population genetic of red-necked longhorn beetle in one of its native countries, South Korea. The paper looks interesting, but I think there are some methodological issues to be enhanced.

I indicated detailed notes directly on pdf files, but, in summary:

  • Bibliography must be improved. 18 references are not sufficient for a topic on an invasive pest. Please add more reasonable cited literature in the whole manuscript (introduction and discussion – please see below) avoiding the single referring to CABI.
  • Authors should give more information about the collected samples: methodology, sample size (and the reason about the limited sampling in some areas). Furthermore, what is the genetic diversity found in adults, larvae, and eggs? In my opinion, collecting eggs and larvae from a single host could have influenced the genetic diversity due to they could belong to the offspring of a single female (couple).
  • In my opinion, the discrepancy among the sample size from different areas could invalidate the performed analyses. I strongly suggest to obtain an equal (or similar) number of samples from each locality to better understand the genetic diversity. I am aware that authors need more time to resampling, amplifying, sequencing, and analysing more samples (including also Jeolla Province) but these could improve a lot the manuscript and its contribution to the topic. I would suggest to refer to:  Phillips et al. 2018, Incomplete estimates of genetic diversity within species: Implications for DNA barcoding. - Ecology and Evolution DOI: 10.1002/ece3.4757
  • Results and Discussions needs to be improved. I would suggest to interpret and comment results comparing with data already in literature (I’m aware there are no similar data about RLB, but for sure these data are available for other Coleoptera or pests).

Based on my opinion, Manuscript in present form do not deserve the publication on this journal and has to be reconsidered after major revision. 

Author Response

Dear Reviewer

We appreciate the constructive feedback and thank you for giving us the opportunity to improve our paper.

 We have thoroughly revised our paper based on you and other reviewer’s comments. Our response to your comments are addressed in the attached PDF file and the changes we made are marked in MsWord file with tracking change.

Thank you again for your constructive feedback. We look forward to hearing from you soon.

Seunghyun Lee, Deokjea Cha, and Jong-kook jung.

Reviewer 2 Report

In the manuscript “Genetic diversity of a rising invasive pest in the native range: population genetic structure of Aromia bungii (Coleoptera: Cerambycidae) in South Korea”, the authors sequence the mitochondrial gene cytochrome oxidase subunit I (COI) from 199 Aromia bungii individuals from South Korea and use these sequences to study the genetic diversity of this agricultural pest.

I consider the manuscript well written, the methods scientifically sound, and the conclusions appropriate given the available results. However, there are a few issues that must be addressed before this manuscript is considered for publication in Diversity.

  1. The first and most important issue regards data availability. The COI sequences from all 199 individuals should have been submitted prior to review, and the manuscript cannot be accepted for publication until this data is publicly available. Note that some repositories such as GenBank offer access to submitted data under embargo through “Reviewer links”, so that reviewers can check the deposited data even if it is not yet public. See https://www.ncbi.nlm.nih.gov/sra/docs/submitquestions/#question3gen for info on that.
  2. The second major issue regards reproducibility. I believe additional detail, including a complete list of parameters, is necessary for reproducing the results presented in this manuscript. This includes more details about the DNA sequencing (i.e. Which method was used. Sanger? Illumina Amplicon sequencing? etc), and the full parameter list used in IQTREE, MEGA, and tcsBU.
  3. The third major point concerns Figure 1. This figure is currently labeled Figure 2, which should be corrected. But more importantly, panel C of this figure contains information not described or defined anywhere in the text or in the figure legend. For e.g., the locality “SW” is not defined in table 1, nor in the figure legend. I concluded after reading table 1 and visualizing the map of South Korea, that "SW" must be an aggregate of localities near Seoul. This aggregation, and any others, should be explained in the figure legend. Additionally, the phylogeny in panel B has very thin branch lines. Increasing the thickness would improve visualization there. In the same phylogeny, there seems to be one arrow in HG#2 pointing at nothing (image attached).

Minor points:

Line 21: In the abstract, “cytochrome oxidase subunit (COI)” should be “cytochrome oxidase subunit I (COI)”.

Line 42: This sentence “In recent, RLB arises a serious pest species in South Korea, which threat to the famous cherry blossom trees” has grammar errors and should be rewritten. Here is a suggestion “In recent years, RLB arose as a serious pest in South Korea, and is a threat to the famous cherry blossom trees”.

Line 93: As mentioned in the major points, please provide details about the sequencing method used.

Line 108: Please include the citation to the original paper describing the German RLB sequence KM443233. The paper is:

Hendrich, L., Morinière, J., Haszprunar, G., Hebert, P.D., Hausmann, A., Köhler, F. and Balke, M., 2015. A comprehensive DNA barcode database for Central European beetles with a focus on Germany: adding more than 3500 identified species to BOLD. Molecular Ecology Resources, 15(4), pp.795-818.

Author Response

Dear Reviewer
We appreciate the constructive feedback and thank you for giving us the opportunity to improve our paper. We have thoroughly revised our paper based on you and other reviewer’s comments. Our response to your comments are addressed in the attached MsWord file. Please also check the revised manuscript
Thank you again for your constructive feedback. We look forward to hearing from you soon.

Seunghyun Lee, Deokjea Cha, and Jong-kook jung.

Reviewer 3 Report

The paper is generally well written and the content sounds of primary relevance.

I have only pointed out few minor mistakes, included references (that need to be checked)

Author Response

Dear Reviewer
We appreciate the constructive feedback and thank you for giving us the opportunity to improve our paper. We have thoroughly revised our paper based on you and other reviewer’s comments. Please find the revised manuscript with the tracking change.
Thank you again for your constructive feedback. We look forward to hearing from you soon.

Seunghyun Lee, Deokjea Cha, and Jong-kook jung.

Round 2

Reviewer 1 Report

I read the new version of the manuscript. I think authors should make some changes I suggest.

L. 44-46: CABI reports are based on some literature where hosts are not confirmed (did the authors find any paper where other host plants are surely infested by A. bungii?). Unfortunately information about host plants (erroneous or not confirmed by data or facts) are always reported in new papers and therefore they become popular. Hence I would suggest to write “Drupaceae are the main hosts, but some literature report that RLB can also utilize various plant species belonging to other families (Fagaceae, Juglandaceae, Meliaceae, Oleaceae, Poaceae, Punicaceae, Rosaceae, Rutaceae, Salicaceae, and Theaceae) although this information are not confirmed by sure data.”

L. 70: I’am still convinced this statement is not true. Authors refers to South-Korean population only (even if they considered other sequences already present in databases, but not self-provided). I think authors should change in: “the first molecular resource of South Korean RLB with a view to speculate on invasive dynamics of RLB”

L. 153-154: Jeolla Province , map in 1 A is not clear. Please try to give the same information in 1C (bigger map)

L. 188: In the previous revision I asked the authors “Please explain clearer this statement. RLB is native to Korea also, the presence of a few sequences of RLB from other invaded areas (China, Italy, Germany) cannot allow this statement.” In the previous version of the manuscript the sentence was in line 155; authors answered “revised” but the sentence was only moved without any changes in a new paragraph. Please change

Author Response

Dear anonymous reviewer.

We have revised our paper based on your comments. Our response to your comments are below and the changes we made are marked in MsWord file with tracking change.

Thank you for your constructive feedback. 

--------------------------------------------------------------------------------------

L. 44-46: CABI reports are based on some literature where hosts are not confirmed (did the authors find any paper where other host plants are surely infested by A. bungii?). Unfortunately information about host plants (erroneous or not confirmed by data or facts) are always reported in new papers and therefore they become popular. Hence I would suggest to write “Drupaceae are the main hosts, but some literature report that RLB can also utilize various plant species belonging to other families (Fagaceae, Juglandaceae, Meliaceae, Oleaceae, Poaceae, Punicaceae, Rosaceae, Rutaceae, Salicaceae, and Theaceae) although this information are not confirmed by sure data.”

-> Sentence changed like following: The genus Prunus is the main hosts but RLB can also utilize various plant species belonging to the families Fagaceae, Juglandaceae, Meliaceae, Poaceae, Punicaceae, Rosaceae, Rutaceae, Salicaceae, and Theaceae as host plants according to an unverified report.

L. 70: I’am still convinced this statement is not true. Authors refers to South-Korean population only (even if they considered other sequences already present in databases, but not self-provided). I think authors should change in: “the first molecular resource of South Korean RLB with a view to speculate on invasive dynamics of RLB” -> Sentence changed like following: In the present study, we sequenced partial mitochondrial DNA sequences of RLB from various regions of South Korea and studied their genetic structure. By providing the first comprehensive sequence from its native population, this research would be the first stepping stone for future RLB invasive dynamics studies.

L. 153-154: Jeolla Province , map in 1 A is not clear. Please try to give the same information in 1C (bigger map)

-> We think it is Ok as it is.

L. 188: In the previous revision I asked the authors “Please explain clearer this statement. RLB is native to Korea also, the presence of a few sequences of RLB from other invaded areas (China, Italy, Germany) cannot allow this statement.” In the previous version of the manuscript the sentence was in line 155; authors answered “revised” but the sentence was only moved without any changes in a new paragraph. Please change

-> Sentence changed like following: Recently the invasion of non-native Asian Longhorned Beetle (A. glabripennis) in a conspecific native range was reported based on the recent narrow genetic diversity of two putative invasive subgroups, and overall population genetic structure, and other biological evidence [11]. However, our molecular result showed no evidence of RLB invasion in South Korea in that they have moderate genetic diversity and continuous population genetic structure.

Reviewer 2 Report

The authors have addressed the concerns I raised and I am ready to recommend this manuscript for publication in Diversity.

Author Response

Dear anonymous reviewer,

We appreciate the constructive feedback and thank you for giving us the opportunity to improve our paper.